# Investigating the optimal handling of uncertain pregnancy episodes in the CPRD GOLD Pregnancy Register: a methodological study using UK primary care data

Jennifer Campbell  ,[1,2] Krishnan Bhaskaran,[2] Sara Thomas,[3] Rachael Williams,[1] Helen I McDonald  ,[3] Caroline Minassian[2]

[1]Clinical Practice Research Datalink, Medicines and Healthcare Products Regulatory Agency, London, UK
[2]Faculty of Epidemiology and Population Health, Department of Non-communicable Disease Epidemiology, London School of Hygiene and Tropical Medicine, London, UK
[3]Faculty of Epidemiology and Population Health, Department of Infectious Disease Epidemiology, London School of Hygiene and Tropical Medicine, London, UK

**Correspondence to**
Jennifer Campbell;
jennifer.campbell@mhra.gov.uk

## ABSTRACT

**Objectives** To investigate why episodes of pregnancy identified from electronic health records may be incomplete or conflicting (overlapping), and provide guidance on how to handle them.

**Setting** Pregnancy Register generated from the Clinical Practice Research Datalink (CPRD) GOLD UK primary care database.

**Participants** Female patients with at least one pregnancy episode in the Register (01 January 1937–31 December 2017) which had no recorded outcome or conflicted with another episode.

**Design** We identified multiple scenarios potentially explaining why uncertain episodes occur. Criteria were established and systematically applied to determine whether episodes had evidence of each scenario. Linked Hospital Episode Statistics were used to identify pregnancy events not captured in primary care.

**Results** Of 5.8 million pregnancy episodes in the Register, 932 604 (16%) had no recorded outcome, and 478 341 (8.5%) conflicted with another episode (251 026 distinct conflicting pairs of episodes among 210 593 women). 826 146 (89%) of the episodes without outcome recorded in primary care and 215 577 (86%) of the conflicting pairs were consistent with one or more of our proposed scenarios. For 689 737 (74%) episodes with recorded outcome missing and 215 544 (86%) of the conflicting pairs (at least one episode), supportive evidence (eg, antenatal records, linked hospital records) suggested they were true and current pregnancies. Furthermore, 516 818 (55 %) and 160 936 (64%), respectively, were during research quality follow-up time. For a sizeable proportion of uncertain episode, there is evidence to suggest that historical outcomes being recorded by the general practitioner during an ongoing pregnancy may offer explanation (73 208 (29.2%) and 349 874 (37.5%)).

**Conclusions** This work provides insight to users of the CPRD Pregnancy Register on why uncertain pregnancy episodes exist and indicates that most of these episodes are likely to be real pregnancies. Guidance is given to help researchers consider whether to include/exclude uncertain pregnancies from their studies, and how to tailor approaches to minimise underestimation and bias.

### Strengths and limitations of this study

► This work carefully examines the way in which pregnancies are recorded in electronic health data in order to maximise its usefulness for pregnancy research.

► Detailed scenarios were developed as to why uncertain pregnancy episodes may occur along with criteria which researchers can apply to ascertain which episodes may fit each scenario.

► Clinician advice and clinical guidelines were used to generate assumptions as to why and when clinicians may record information relating to pregnancy; however, these may not be correct in every case.

► Electronic health data are not collected for the purposes of research and can be messy for a variety of reasons, some of which may not have been captured in this study.

## INTRODUCTION

Understanding how diseases, drugs and other exposures affect pregnant women and their children is an important public health priority. However, pregnant women are excluded from many trials due to potential risks to the woman and her unborn child. Observational research using electronic healthcare records (EHRs) has thus become a well-established vital tool for investigating disease prevalence, risk factors and pharmacovigilance in pregnant women. UK primary care databases are particularly useful due to the gate-keeper healthcare system meaning all antenatal care is overseen by a general practitioner (GP).[1] One example of such a database is CPRD GOLD. This database is produced and maintained by the Clinical Practice Research Datalink (CPRD), a government research service collecting de-identified and fully coded patient-level

BMJ

EHR from primary care practices across the UK.[2] However, challenges such as incomplete data capture in EHR data can make it difficult to identify accurately the start and end of pregnancies. Recently, a collaboration between CPRD and the London School of Hygiene and Tropical Medicine established a Pregnancy Register of all pregnancies in CPRD GOLD[3] which includes approximately 6 million estimated pregnancies (henceforth, pregnancies in the Register will be referred to as pregnancy episodes).

Previous approaches to generating pregnancy registers have been limited by the exclusion of pregnancies without identified outcomes and pregnancy records which do not fit chronologically into an identified pregnancy episode.[4] Ignoring these records potentially excludes periods when women were pregnant. If these pregnancies systematically differ from those captured more completely, their exclusion may lead to bias. For example, pregnancies ending in miscarriage may be less likely to have the outcome recorded than pregnancies ending in live birth.[3] Ignoring pregnancy data which are challenging to interpret may therefore underestimate adverse outcomes. Incomplete capture of pregnancies also impacts descriptive studies that need pregnancies as denominator data, such as vaccine uptake studies. A further limitation of previous approaches is that some women have pregnancies that seemingly overlap in the data, and these are not addressed. These conflicting pregnancies highlight that estimated timings of some pregnancies may be suboptimal and/or some pregnancy episodes may not be true pregnancies. Approaches which exclude incongruent or incomplete pregnancy data may lead to misclassification of exposure timings.

The unique advantage of the CPRD Pregnancy Register is that it uses all pregnancy data in CPRD GOLD, thereby capturing all documented pregnancies regardless of completeness. However, this also presents interpretational challenges: approximately 950000 pregnancy episodes (16% of all pregnancy episodes) have no outcome recorded and approximately 500000 pregnancy episodes conflict with another episode for the same woman (episodes identified by the algorithm with at least 1 day of overlap). These episodes are flagged in the Register enabling researchers to identify them when designing their study. However, there may be multiple reasons for the occurrence of uncertain episodes and therefore absolute rules on whether to include or exclude them from a study may be inappropriate.

We therefore aimed to investigate possible reasons why the algorithm used to generate the CPRD Pregnancy Register identifies uncertain episodes and thus generate information to guide future use of this important resource. Our specific objectives were:

1. To identify potential scenarios which may result in pregnancy episodes without a recorded outcome or those which conflict with another episode for the same woman.

2. To use available data (including linked data) to investigate these potential scenarios and flag pregnancy episodes which are consistent with each one.

3. To provide information to researchers using the Register to help inform their decisions on how to handle these uncertain episodes when designing studies.

## METHODS
### Data sources
#### CPRD primary care data and the Pregnancy Register
The CPRD GOLD UK primary care database contains registration information and all care events that general practice staff record to support clinical care. This includes demographic information (birth year, sex, etc), clinical events (signs, symptoms, medical diagnoses), referrals to specialists and secondary care, prescriptions issued in primary care, vaccinations, test results, lifestyle information (eg, smoking status) and other care administered as part of GP practice.[5] CPRD data also contain indicators of data quality at the patient level (known as the acceptability flag; online supplemental appendix 1) and at the practice level (known as the practice up-to-standard (UTS) date; online supplemental appendix 1). As CPRD GOLD is a longitudinal database, updated monthly, it contains variables indicating whether the patient and practice are still contributing data.

The Pregnancy Register lists and characterises all pregnancies identified in CPRD GOLD based on an algorithm.[3] A single record represents a unique pregnancy episode. Each woman may have multiple episodes. Information includes the estimated start and end of pregnancy, its outcome (when recorded) and whether it was a singleton or multiple pregnancy. For live birth pregnancies, patient identifiers of linked babies identified through the CPRD Mother-Baby-Link[6] are provided. Figure 1 gives an overview of the algorithm steps, including how gestational ages were applied, and online supplemental appendix 2 gives a list of the variables provided in the Register. Figure in online supplemental appendix 3 shows an example of how a real pregnancy might manifest in (a) raw CPRD gold data and (b) the processed Pregnancy Register dataset.

#### Linked data
Person-level linkage of CPRD primary care data with other datasets (eg, Hospital Episode Statistics HES) is available for English practices who have consented to participate in the linkage scheme.[7] These linkages cover approximately ~56% of contributing CPRD GOLD practices in the UK. Where available, we used linked data to look for further information about the pregnancy episodes within the Register. HES APC (Admitted Patient Care) data include information on admission and discharge dates, diagnoses, specialists seen and procedures undertaken for linked patients with a hospitalisation record.[8] We searched HES APC data for records of pregnancy outcomes using International Classification of Diseases (ICD-10) and

**1. Identify all pregnancy outcome records** : All records relating to pregnancy outcomes of any type (live births, stillbirths and early pregnancy losses) are extracted. Records relating to deliveries are considered separately to those relating to early pregnancy losses.

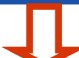

**2. Date the first pregnancy outcome:** The date of each woman's first pregnancy outcome is estimated using the records identified in step 1 and additional data from linked babies' records (for live births, when available) .

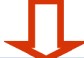

**3. Group together records relating to the first pregnancy outcome and  characterise the pregnancy outcome:** Additional pregnancy outcome records relating to each woman's first outcome which are <= 25 weeks apart  for deliveries and <= 8weeks apart for pregnancy losses, are assigned to that outcome. Characteristics of the delivery (pre- or post-term, stillbirth, multiple birth) or the type of early pregnancy loss (miscarriage, induced abortion, ectopic, molar pregnancy or blighted ovum) are determined from these assigned records.

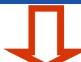

**4. Date and characterise each successive pregnancy outcome** : Steps 2 & 3 are repeated to identify, date and characterise successive pregnancy outcomes sequentially for each woman.

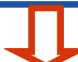

**5. Estimate the start of each pregnancy episode** : Records relating to the timing of the start of pregnancy (first day of last menstrual period) are used to estimate pregnancy start dates. These include information such as records of gestational age, estimated date of delivery and estimated date of conception. In the absence of such data, pregnancy start dates are imputed according to the type of pregnancy outcome  (40 weeks when the pregnancy is not flagged as preterm, post-term or multiple , 36 weeks for pre-term pregnancies , 37 weeks for multiple pregnancies, 41 weeks for post term pregnancies).

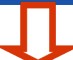

**6. Adjust the start and end dates of a pregnancy** : Adjustments to pregnancy start and end dates are made either when antenatal records are identified in the 4 weeks before the estimated pregnancy start date (indicating that the initial estimated start date was too late), or when the estimated pregnancy duration exceeds the maximum duration for that type of pregnancy outcome.

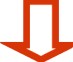

**7. Assign antenatal records to each pregnancy episode** : Antenatal records occurring between the start and end date of an identified pregnancy episode are assigned to the pregnancy.

**8. Identify additional pregnancies with no recorded outcome** : All remaining <u>unassigned</u> antenatal records are extracted and categorised into distinct pregnancy episodes. Successive records which are <= 6 weeks apart are grouped together to create a pregnancy. The date of the latest antenatal record in the episode is used as a proxy for the pregnancy end date. The pregnancy start date is estimated using information on gestational age (when available) or by subtracting 4 weeks from the earliest antenatal record in the episode.

**Figure 1**    Pregnancy register algorithm steps used to create the CPRD Pregnancy Register. CPRD, Clinical Practice Research Datalink.

Operating Procedure Codes (OPCS) (online supplemental appendices 4 and 5). HES APC maternity records were also used: a recording of an acceptable value in any of the variables identified as relating to delivery (online supplemental appendix 6) was taken as evidence that a delivery had taken place.

The HES Diagnostic Imaging Dataset (DID) provides detailed information about diagnostic imaging tests, including X-rays, MRI scans and fetal growth scans, taken from National Health Service (NHS) providers' radiological information systems. This was used for records of fetal scans. Office for National Statistics (ONS) mortality data were also used to ascertain additional death records which may have been missing from CPRD.

We used set 17 of the CPRD linked data for which the coverage periods were: HES APC 01 April 1997–31 July 2017; HES DID 01 April 2012–31 July 2017; ONS Mortality Data 02 January 1998–19 September 2017.

### Study population

This study included all individuals who had at least one pregnancy episode without a recorded outcome or at least one conflicting pregnancy episode in the February 2018 version of the Pregnancy Register. All pregnancy records for these patients were extracted from the CPRD GOLD database using the pregnancy code-list upon which the pregnancy algorithm is based,[3] thereby creating a dataset which included all pregnancy records and the summary Pregnancy Register information for these women. Women were followed up until the minimum of leaving the practice, death or practice last collection date. In the linked data analysis, women with HES records beyond this point were followed up until the end of linked data coverage.

### Identifying scenarios to explain the occurrence of uncertain episodes

Potential scenarios which may result in uncertain pregnancy episodes, including those without recorded outcomes and those which conflicted with another episode, were identified through discussions with the creators of the Register (CM, ST, RW), clinicians and CPRD data experts. The scenarios are based on the structure of the CPRD GOLD data and the Pregnancy Register algorithm (figure 1, steps 1–8). The scenarios are not mutually exclusive; thus, episodes may be consistent with more than one scenario.

### Pregnancy episodes with recorded outcome missing

Scenarios with the potential to result in episodes with missing outcomes were identified. There are four overarching problems with various specific scenarios within them: the pregnancies are true and current, but the outcome was not captured in CPRD primary care data; the pregnancies are true and current, but the pregnancy was still ongoing at the end of follow-up in the database; the patient was not pregnant at the time of the database record; the pregnancy is really part of another pregnancy

episode in the Register. The 12 scenarios which fall under these problems are described in table 1.

### Conflicting pregnancy episodes

Scenarios with the potential to result in conflicting episodes were proposed and are described in detail in table 2. Identifying the scenarios was an iterative process, after applying initial scenarios we took a sample of 50 conflicting pregnancy episodes and reviewed the patient data. This allowed us to validate existing scenarios and identify further scenarios. Scenarios can be grouped under four overarching problems: both pregnancies are true but one is a historical pregnancy; both pregnancies are historical; both pregnancies are true and current but the gestation of the second pregnancy estimated by the algorithm is too long; the woman was pregnant, but one pregnancy has been split into multiple episodes by the rules of the algorithm (online supplemental appendix 3).

### Applying criteria to identify evidence of each scenario
#### Evidence in HES

For each episode, it was ascertained whether the woman was eligible for linkage to other data and whether the episode occurred within the coverage period of each linked data source. For pregnancy episodes occurring within the linkage coverage period, the linked HES data were examined for evidence of pregnancy outcomes. The period for which outcomes were searched was from the episode start date to 9 months after the episode end date; we excluded from this analysis pregnancies where this period was entirely outside the coverage dates for linked HES data.

ICD-10 and OPCS code lists were used to look for evidence of outcomes in the HES APC Episodes, Diagnosis and Procedures tables (online supplemental appendices 4 and 5). In the HES APC maternity data, a recording of an acceptable value in any of the variables identified as relating to delivery (online supplemental appendix 6) was flagged as evidence that a delivery had taken place. In the HES outpatient data, an ICD-10 code list for evidence of delivery, termination or early pregnancy loss was used. Snomed codes (online supplemental appendix 14) were used to identify all fetal scan records in the HES DID data.

### Pregnancy episodes with recorded outcome missing

All episodes coded as outcome unknown ('13' in the outcome field) were extracted from the Pregnancy Register. For each episode, we extracted information on the timing of the episode in relation to the start and end of patient follow-up and the period of research standard (UTS) data recording in CPRD, and we also searched for relevant codes in the patient's record, namely: early pregnancy codes which were likely to be recorded in the patient's first antenatal visits to the GP; codes which are likely to be recorded by the GP as clinically important in the patient's medical history even when the patient was not pregnant; codes which may indicate an outcome but were originally classified by the Register as antenatal;

**Table 1** Description of potential scenarios leading to pregnancy episodes with no recorded outcome and scenario criteria applied

| Scenario | How does this appear in the data? | Criteria used to determine if there is evidence in the data that an episode is consistent with the scenario in question |
|---|---|---|
| **Problem 1:** The women was pregnant at the time of the database record, but the outcome *was not captured in CPRD primary care data.* | | |
| 1a. The woman was pregnant. She had a delivery, miscarriage or termination of pregnancy (TOP) in hospital or elsewhere and information either was not fed back to the general practice, or was fed back but not coded in the woman's records. | There will be no evidence of an outcome in CPRD data up to 38 weeks* (for delivery) or up to 20 weeks (for miscarriage or TOP) after the first antenatal record for the pregnancy. However, there may be evidence of delivery/miscarriage/TOP in one of the linked HES APC data. | ► The woman must be eligible for linkage.<br>► There must be at least 1 day of overlap between the data coverage for each HES source and the pregstart+294 days (42 weeks) to give a maximum potential end date.<br>► There must be a record in HES of delivery or loss within 294 days (42 weeks). |
| 1b. The pregnancy outcome was recorded in the primary care data but has no event date recorded alongside it and is therefore not picked up by the algorithm. | There will be an outcome code with missing eventdate† within 38 weeks after the first antenatal record of the pregnancy episode (using the systemdate† as a proxy for the event date). | ► There must be an antenatal code with missing eventdate† recorded with a systemdate† ≥294 days after pregnancy episode starts. |
| 1c. The pregnancy outcome occurred before the patient was registered at their current practice or before the start of the practice up-to-standard follow-up (UTS). When the patient joined the practice, information was recorded about the pregnancy but not the outcome. | The pregnancy episode will occur before the start of the patient's current registration and/or UTS. | ► Pregnancy episode end date must be <UTS date† OR ≤current registration date. |
| **Problem 2:** The women was pregnant at the time of the database record*, but the pregnancy was still ongoing at the end of available follow-up in the database.* | | |
| 2a. The woman moved practices before the end of her pregnancy. If a patient transfers out of a CPRD practice, then follow-up is lost. OR The woman died before the end of her pregnancy. | There will be a transfer out date or death date (in either CPRD or the ONS mortality data) less than 38 weeks after the earliest antenatal record for the pregnancy episode. | ► The earliest of the woman's transfer out date† or death date (in either CPRD or the ONS mortality data) minus pregnancy episode start date must be ≤294 days. |
| 2b. The last collection of data from the practice was before the pregnancy outcome. | There will be a last collection date less than 42 weeks after the start of the pregnancy episode. | ► The woman's last collection date minus pregnancy episode start date must be ≤294 days. |
| **Problem 3:** *The patient was not pregnant at the time of the database record.* | | |
| 3a. A historical pregnancy was recorded retrospectively in the first few months after patient joins the practice. In this scenario, information about the pregnancy is recorded with the current date (by GP software default) rather than the date it occurred (different from scenario 1c). This is more likely to occur when a woman joins a practice and the GP may wish to record past pregnancy events which are relevant to her current clinical care. | The pregnancy episode will occur less than 1 year after the women's current registration date. There will be a record of a pregnancy event which may be clinically useful for future care between the start and end of the pregnancy episode. | ► Pregnancy episode start date is <365 days after current registration date.<br>► There is a record of a pregnancy code from a list identified as likely to be recorded as useful pregnancy history information (online supplemental appendix 7).<br>► This must have an eventdate ≥pregstart† & ≤pregend.† |
| 3b. The woman was not pregnant but was planning a pregnancy and discussed this with the GP, for example, due to other medical conditions which may complicate pregnancy. | The pregnancy episode will include a pregnancy advice code, for example, '67AF.00 Pregnancy advice for patients with epilepsy'. | ► The woman has antenatal codes identified as pregnancy advice codes (online supplemental appendix 8) with an eventdate† ≥pregstart† & ≤pregend.† |
| **Problem 4:** *The pregnancy record belongs to another pregnancy episode in the Register.* | | |
| 4a. There was a delay in recording the outcome of a pregnancy by the practice. Thus, the outcome code has an eventdate† which is later than the true outcome date. The algorithm then calculates the Last Menstrual Period (LMP) date as being later than it was (figure 1, steps 5 and 6). Records which occurred early in pregnancy are then left unassigned to the pregnancy episode and appear as if belonging to a previous pregnancy episode which has no outcome recorded (figure 1, step 8). | As the pregnancy episode without outcome has been created from unassigned records at the beginning of the pregnancy, it will be followed by another pregnancy episode. There is unlikely to be more than a 3-month delay in outcome recording due to the mother attending the practice for postnatal checks and/or infant vaccinations. Therefore, there will be less than 12 weeks between the end of the episode with no recorded outcome and the start of the next pregnancy episode. | ► The woman must have >1 episode in the Pregnancy Register.<br>► Episodes with recorded outcome missing were eligible if they were not the last pregnancy episode for that woman.<br>► There must be ≤84 days (12 weeks) between the pregend† of the episode without outcome and the pregstart† of the woman's next episode. |
| 4b. The LMP is derived from information in the data and is estimated by the algorithm to have occurred later than reality (figure 1, steps 5). This may lead to a short pregnancy episode and unassigned codes before the estimated start of pregnancy. These are then grouped to form a pregnancy episode with no recorded outcome (figure 1, step 8). | The pregnancy episode without outcome will be followed by another pregnancy episode which will be less than 40 weeks long. | ► The woman must have >1 episode in the Pregnancy Register.<br>► The episode after the episode with missing outcome must have a startsource†=2, 4, 5 or 6 (online supplemental appendix 2). The length (gestdays) of the episode must be <280 days. |

Continued

**Table 1** Continued

| Scenario | How does this appear in the data? | Criteria used to determine if there is evidence in the data that an episode is consistent with the scenario in question |
|---|---|---|
| 4c. If there are pregnancy records within 4 weeks before the estimated LMP, the identified pregnancy episode is shifted earlier in time by the algorithm (within plausible limits) to encompass those records (figure 1, step 6). This may leave unassigned pregnancy records which occurred shortly after the new estimated delivery date which will then be grouped to form a pregnancy episode with no recorded outcome (figure 1, step 8). | The pregnancy episode must not be the only pregnancy for this to apply. There will be another pregnancy episode which ends <8 weeks before the first antenatal record of the pregnancy episode without outcome for which the end has been adjusted by the algorithm. | ▶ The woman must have >1 episode in the Pregnancy Register.<br>▶ The episode before the one with recorded outcome missing must have an endadj†=2 (online supplemental appendix 2).<br>▶ The pregend† date for the episode with missing outcome must be ≤56 days (8 weeks) after the pregend† for that previous episode. |
| 4d. The GP records a code relating to the patient's pregnancy outcome history while the patient is pregnant. This is incorrectly identified by the algorithm as the outcome of the current pregnancy (figure 1, step 3). If the actual outcome is ≤25 weeks after for delivery or ≤12 weeks after for pregnancy losses, they will be grouped together as the same outcome. Subsequent antenatal records may then be grouped together to form a new pregnancy episode with no recorded outcome (figure 1, step 8). | The pregnancy episode must not be the patient's first pregnancy. The pregnancy episode would be within 25 weeks after the previous outcome. | ▶ The woman must have >1 episode in the Pregnancy Register.<br>▶ The pregend† date for the episode with missing outcome had to be ≤175 days (25 weeks) after the pregend† for the previous episode. |
| 4e. The outcome of the pregnancy episode has been misclassified as an antenatal event, for example, 'Failed abortion', 'refer to TOP counselling', 'premature labour', etc. | There will be an antenatal code which should have been an outcome code within 38 weeks after the first antenatal record of the pregnancy episode with recorded outcome missing. | ▶ There must be an antenatal record from a code list of potentially misclassified outcomes (online supplemental appendix 9) 266 days (38 weeks) of the firstantenatal† record. |

*The first antenatal record is assumed to be recorded ≥4 weeks after the LMP as the woman is unlikely to know she is pregnant before then.
†Refers to a CPRD GOLD-specific variable, for example: pregend=the end of episode as defined by the algorithm; pregstart=the start of episode as defined by the algorithm; endadj=an indication that the end of the episode has been adjusted and how; startsource=which data were used to generate the start of the episode. These variables and others are defined in more detail in online supplemental appendix 2.
APC, Admitted Patient Care; CPRD, Clinical Practice Research Datalink; GP, general practitioner; HES, Hospital Episode Statistics; ONS, Office for National Statistics.

codes which are likely to be recorded by the GP as part of a consultation about the potential health impacts on a patient of becoming pregnant (code lists in online supplemental appendices 7–9).

For each scenario, a set of criteria based on how these should appear in the data were established (described in detail in table 1). Criteria were systematically applied to the data to establish which episodes were consistent with each scenario.

### Conflicting pregnancy episodes

All conflicting episodes (those with at least 1 day of overlap with another episode for the same woman) were ascertained using the conflict flag in the Register. Pregnancy episodes may conflict with more than one other episode. Each conflicting pair was treated separately and therefore an individual pregnancy episode could appear in the analysis multiple times. A dataset was created which contained one row per pair of conflicting pregnancy episodes.

Episodes were ordered by start date with episode one being the earlier start date of the two. Descriptive variables were added to the dataset from the CPRD GOLD data to indicate if the episodes were during current registration and UTS follow-up. Pregnancy episode outcomes were grouped into three categories: delivery, loss or missing, and a variable was generated to indicate the combination of outcomes in each conflicting pair (online supplemental appendix 12).

For each scenario, a set of criteria based on how these should appear in the data were established (described in detail in table 2). Criteria were systematically applied to the data to establish which conflicting pairs were consistent with each scenario.

### PATIENT AND PUBLIC INVOLVEMENT

There was no patient or public involvement in this methodological work.

### RESULTS

There were 2 438 493 women with a pregnancy episode in the February 2018 version of the Pregnancy Register; of these patients, 731 368 (30%) had at least one uncertain episode. Mean patient follow-up time for all women was 4720 days, this was slightly lower for women with a missing outcome record (4349 days) (table 2). Women with an uncertain episode were more likely to be over 30 years of age. Uncertain pregnancy episodes were also more likely to be recent (after 2000) (table 2).

### Pregnancy episodes with recorded outcome missing

Of the 5.8 million pregnancy episodes in the Pregnancy Register, there were 932 604 (16%) episodes with no recorded outcome of which over half (516 818, 55.4%) were during UTS follow-up and current registration (table 3). A total of 826 146 (89%) had evidence

 Campbell J, et al. BMJ Open 2022;12:e055773. doi:10.1136/bmjopen-2021-055773

**Table 2** Description of potential scenarios leading to conflicting episodes and scenario criteria applied

| Scenario | How does this appear in the data? | Criteria applied to pairs of conflicting episodes to determine if there is evidence in the data that the pair is consistent with the scenario in question |
|---|---|---|
| **Problem 1:** *Both pregnancies are true, but one is a current pregnancy and one is a historical pregnancy.* | | |
| 1a. The GP records a past delivery during a current pregnancy >25 weeks before the true delivery of that pregnancy. OR a past pregnancy loss >12 weeks before the actual loss of that pregnancy. | Both pregnancies will have the same outcome type. Evidence of current pregnancy codes would be expected to fall within the second pregnancy. | ► The outcome combination of the two episodes must be delivery/delivery or loss/loss (see online supplemental appendix 10 for outcome classifications).<br>► The second episode had an antenatal code from a list deemed likely to only be recorded if the patient was currently pregnant (online supplemental appendix 11) OR a scan record in the HES DID data between firstantenatal* and pregend*. |
| 1b. If a patient has a record relating to a previous loss recorded during a pregnancy ending in delivery or vice-versa, then conflicting episodes will be created by the algorithm. The algorithm first generates episodes for consecutive deliveries; it then does the same thing for pregnancy losses. There is no step in the algorithm to check that the loss episodes do not coincide with the delivery episodes (figure 1, steps 1–6). | The conflicting pregnancies must consist of one loss and one delivery. Evidence of current pregnancy codes would be expected to fall within the second pregnancy. | ► The outcome combination of the two episodes must be delivery/loss or loss/delivery (see online supplemental appendix 10 for outcome classifications).<br>► The second episode had an antenatal code from a list deemed likely to only be recorded if the patient was currently pregnant (online supplemental appendix 11) OR an antenatal scan record in the HES DID data between firstantenatal* and pregend.* |
| **Problem 2:** *Both pregnancies are historical.* | | |
| 2a. A patient joins a new practice (or has another reason for a full obstetric history to be taken) and has information on historical pregnancies recorded with the current date rather than the actual date of the event. Losses and deliveries recorded on the same date will result in conflicting episodes in the Register as different outcome types are generated separately by the algorithm (figure 1, steps 1–5). | The conflicting pregnancies must consist of one loss and one delivery. The pregnancy end dates will be the same for both pregnancies. Both pregnancies are likely to be <1 year after the patient's current registration date. We would not expect to find codes indicating current pregnancy. | ► The outcome combination of the two episodes must be a delivery and a loss.<br>► The pregend* dates must be the same.<br>► There must be no antenatal codes relating to current pregnancy (online supplemental appendix 11) or HES DID antenatal scan recorded between the firstantenatal* date and the pregend* date of either episode. |
| **Problem 3:** *Both pregnancies are true and current but the gestation of the second pregnancy estimated by the algorithm is too long.* | | |
| 3a. The woman has two pregnancy losses which are >8 weeks and <12 weeks apart. The second pregnancy has no information about gestation recorded so the algorithm applies a default of 12 weeks and the episodes overlap. | Both conflicting pregnancies must be losses. The maximum overlap between the two pregnancies must be 4 weeks. Evidence of current pregnancy codes could be found in either pregnancy. | ► The outcome combination of the two episodes must be two losses. The pregend* for the first episode must be ≤28 days after the pregstart* of the second episode. |
| 3b. The woman has two pregnancies close together and the second pregnancy ends in delivery. If the information on the Last Menstrual Period date (LMP) in the data of the second pregnancy is wrong, then the algorithm may generate the start too early resulting in an overlap. | The second pregnancy must be a delivery and have no information about gestation in the data. The overlap must be <15 weeks (otherwise the two outcomes would be <25 weeks apart and would have been grouped as one; see figure 1, step 3). There may be evidence of current pregnancy codes in either pregnancy. | ► The outcome of the second episode must be a delivery.<br>► The startsource* of the second episode must not be equal to 4 or 5 (online supplemental appendix 2).<br>► The pregstart* of the second episode must be <105 days (15 weeks) before the pregend* of the first episode. |
| **Problem 4:** *The pregnancy is true and current but is split into separate episodes by the rules of the algorithm.* | | |
| 4a. The GP records further information about a pregnancy outcome >25 weeks after the delivery date for pregnancies ending in delivery OR >8 weeks but <12 weeks for pregnancies ending in loss. The algorithm assumes this further information is a different pregnancy and generates a new episode, which may overlap with the 'true' episode. | Both pregnancies must be of the same outcome type. Evidence of current pregnancy codes would be expected to fall within the first pregnancy. | ► The outcome combination of the two episodes must be delivery/delivery or loss/loss (online supplemental appendix 12).<br>► The first episode had an antenatal code from a list deemed likely to only be recorded if the patient was currently pregnant (online supplemental appendix 11) OR a scan record in the HES DID data between firstantenatal* and pregend*. |

**Table 2**    Continued

| Scenario | How does this appear in the data? | Criteria applied to pairs of conflicting episodes to determine if there is evidence in the data that the pair is consistent with the scenario in question |
|---|---|---|
| 4b. The GP records further antenatal information about a pregnancy after delivery or pregnancy loss. This will then be used to generate a new pregnancy without outcome episode by the algorithm. If the code is within 4 weeks of the end of the true pregnancy episode, the two will overlap. | The first pregnancy must be a pregnancy with an outcome recorded in the data. The second pregnancy must be a pregnancy without outcome which consists of one antenatal code not related to a scan. | ► The first episode must have outcome=1–10 in the Register (online supplemental appendix 2) and must have endadj*=0.<br>► The second episode must have no recorded outcome (outcome=13).<br>► The second episode must have a gestdays*=28 (likely to consist of one code) and there must NOT be a scan code (online supplemental appendix 13) with an eventdate*=pregend* of the second episode. |
| 4c. The patient has a follow-up scan after a pregnancy loss. This is recorded in the data by the GP as an antenatal scan. The algorithm then creates a second pregnancy episode based on the antenatal scan code which becomes a pregnancy without outcome in the Register. | The first pregnancy must be a pregnancy loss. The second pregnancy must be a pregnancy without outcome which consists of one antenatal code related to a scan. | ► The outcome combination of the two episodes must be loss/missing.<br>► The second episode must have a gestdays*=28 (likely to consist of one code) and there must be a scan code (online supplemental appendix 13) with an eventdate*=pregend* of the second episode. |
| 4d. The GP records information about a pregnancy but no information about the outcome. If records relating to this pregnancy are more than 6 weeks apart, they will be turned into multiple episodes. Once estimated start dates are generated for these episodes based on the data recorded (figure 1, step 8), episodes may overlap. For example, if there is gestational information included in the second episode, the start of this episode will be assigned before the start of the previous episode resulting in a nested pregnancy episode. | Both pregnancies must be pregnancies without outcome in the Register. The end of the first pregnancy must be greater than 6 weeks before the first antenatal of the second. | ► The outcome combination of the two episodes must be missing/missing.<br>► The pregend* of the first episode is >42 days before the firstantenatal* date of the second episode. |
| 4e. The first pregnancy episode ended in delivery and has been shifted backwards by the rules of the algorithm leaving unassigned late pregnancy or third trimester records. These records will then be identified by the algorithm as end of pregnancies (figure 1, step 6) and new conflicting episodes will be created. | The first pregnancy must be a pregnancy with a delivery outcome recorded in the data. The end of the first pregnancy must have been adjusted. The second pregnancy must be a pregnancy where the outcome is based on a late pregnancy or third trimester record. | ► The first episode must have a delivery outcome code and endadj* variable not=0.<br>► The second episode must have outcome=11, 12 or 13. |

*Refers to a CPRD GOLD-specific variable, for example: pregend=the end of episode as defined by the algorithm; pregstart=the start of episode as defined by the algorithm; endadj=an indication that the end of the episode has been adjusted and how; startsource=which data were used to generate the start of the episode. These variables and others are defined in more detail in online supplemental appendix 2.
CPRD, Clinical Practice Research Datalink; DID, Diagnostic Imaging Dataset; GP, general practitioner; HES, Hospital Episode Statistics.

consistent with at least one of the identified scenarios (table 4). On the other hand, 689 737 (74%) had evidence of a scenario indicating they were true (either current or historical) pregnancies (scenarios 1a, 1b, 1c, 2a, 2b or 4e). The largest proportion of pregnancy episodes occurred before the patient registered at their current practice which contributed the data to CPRD or before that practice was deemed to be contributing research standard data (415 807, 44.6% scenario 1c). A total of 211 070 (22.6%) episodes had data in HES consistent with the outcome occurring in hospital and not being fed back to the GP (scenario 1a), representing approximately 50% of episodes with recorded outcome missing which were eligible for linkage. HES APC data were the most useful linked data source for ascertaining pregnancy outcomes with a small number found in HES outpatient (online supplemental appendix 15).

The second most common potential explanation for pregnancies without outcome was scenario 4d, where a code relating to the patient's pregnancy history may have been recorded by the GP while the patient was pregnant. A total of 349 874 (37.5%) episodes without outcome were consistent with this scenario. Relatively fewer episodes were consistent with scenario 4a, 4b and 4e, none were consistent with 4c. For 242 698 (26%) episodes, follow-up ended before the predicted end of the pregnancy (scenario 2a and 2b) for 822 episodes (<0.1%) of these episodes follow-up ended due to death. Only small proportions of episodes were consistent with other scenarios. The distribution of scenarios that occurred during the period left censored by the practice UTS date and patient current registration date was similar to that of the Pregnancy Register as a whole (table 4, online supplemental appendix 16).

**Table 3** Baseline characteristics of the pregnancy episodes in the February 2018 Pregnancy Register

| | Episodes with recorded outcome missing N (%) | Conflicting episodes N (%) | All episodes in the Pregnancy Register N (%) |
|---|---|---|---|
| *Number of patients* | *643 689 (26.4)* | *210 593 (8.6)* | *2 438 493* |
| *Mean patient follow-up time (years)* | *11.92* | *12.92* | *12.93* |
| *Mean number of pregnancy episodes per patient* | *3.63* | *4.66* | *3.44* |
| Pregnancy end was during UTS follow-up and current registration | 516 818 (55.4) | 160 936 (64.1) | 1 926 077 (33.1) |
| Age group of the patient at the end of the pregnancy episode | | | |
| 11–14 | 1344 (0.1) | 76 (0.0) | 7867 (0.1) |
| 15–19 | 72 543 (7.8) | 15 420 (6.1) | 551 025 (9.5) |
| 20–24 | 196 979 (21.1) | 48 273 (19.2) | 1 397 717 (24.0) |
| 25–29 | 254 352 (27.3) | 65 601 (26.1) | 1 624 350 (27.9) |
| 30–34 | 235 995 (25.3) | 69 236 (27.6) | 1 339 439 (23.0) |
| 35–39 | 126 369 (13.6) | 40 079 (16.0) | 685 421 (11.8) |
| 40–44 | 37 640 (4.0) | 11 355 (4.5) | 194 354 (3.3) |
| 45–49 | 7382 (0.8) | 953 (0.4) | 24 208 (0.4) |
| Year pregnancy episode ended | | | |
| pre-1950 | 1417 (0.2) | 41 (0.0) | 16 695 (0.3) |
| 1950–1959 | 8061 (0.9) | 522 (0.2) | 98 436 (1.7) |
| 1960–1969 | 19 312 (2.1) | 1887 (0.8) | 283 757 (4.9) |
| 1970–1979 | 24 296 (2.6) | 3882 (1.5) | 493 217 (8.5) |
| 1980–1989 | 38 768 (4.2) | 9135 (3.6) | 803 380 (13.8) |
| 1990–1999 | 248 016 (26.6) | 54 254 (21.6) | 1 530 212 (26.3) |
| 2000–2009 | 336 523 (36.1) | 116 429 (46.4) | 1 705 380 (29.3) |
| 2010–2018 | 256 211 (27.5) | 64 843 (25.8) | 893 304 (15.3) |
| **Total pregnancies** | 932 604 | 251 026 | 5 824 381 |

UTS, up-to-standard.

## Conflicting pregnancy episodes

There were 478 341 (8.5%) pregnancy episodes with a conflict recorded in the February 2018 Pregnancy Register, amounting to 251 026 conflicting pregnancy pairs. Over half of the pairs (160 936, 64%) were during UTS follow-up and current registration. There were 215 577 (88.6%) pairs which were consistent with at least one identified scenario. Of the remaining 106 458 (11.4%), less than half were during UTS follow-up and current registration (table showing these pregnancies by scenario is given in online supplemental appendix 17). Across all scenarios, at least 40% were during UTS follow-up and current registration. Of the pregnancy pairs, 215 544 (86%) evidence of a scenario indicating that at least one episode was a true and current pregnancy (scenarios 1a, 1b, 3a, 3b and 4a–e). Most conflicting pairs had at least one pregnancy episode ending in loss (201 783, 80.3%) (online supplemental appendix 18). Furthermore, 41% (101 760) of pairs included at least one pregnancy with no outcome recorded.

A total of 75 672 (30%) of all conflicting pairs were shown to have evidence that they were consistent with problem 1, that a patient had a record relating to the outcome of a previous pregnancy recorded during a current pregnancy. This includes scenario 1b: a record of a previous loss recorded during a pregnancy ending in delivery or vice-versa, one of the most common scenarios (29% of conflicting pairs) (table 5).

A total of 73 191 (29%) of pairs were consistent with scenario 4e: that adjusting of pregnancy dates by the algorithm had led to unassigned records. Of these, over 96% (70 472) were consistent with this scenario only, and 73% (53 464) of these pairs had a linked baby identified. A total of 43 581 (17.4%) of episodes had evidence that they were consistent with further antenatal information having been recorded after the end of pregnancy (scenario 4b).

For approximately 16% (39,373) of conflicting pairs, there was evidence to suggest that the gestation of the second pregnancy episode specified by the algorithm may

**Table 4** Numbers of pregnancy episodes with recorded outcome missing which were consistent with applied criteria for each scenario*

| Scenario | Description | N of pregnancy episodes with evidence of this scenario (% of total episodes with missing outcome) | N of pregnancy episodes with evidence of this scenario only (% of total episodes with missing outcome) | N of pregnancy episodes with evidence of an outcome in linked HES (% of linkage eligible episodes with recorded outcome missing†) | N of episodes during current registration and UTS follow-up (% of total episodes with missing outcome)* |
|---|---|---|---|---|---|
| Denominator | | 932 604 | 932 604 | 424 375† | 932 604 |
| *Problem 1: The women was pregnant at the time of the database record, but the outcome was not captured in CPRD primary care data.* | | | | | |
| Scenario 1a | The pregnancy outcome occurred in hospital or elsewhere and information wasn't fed back to the practice. | 211 070 (22.6) | 1934 (0.2) | 211 070 (49.7) | 139 084 (14.9) |
| Scenario 1b | The outcome of the pregnancy is recorded in the primary care data but has no event date associated with it. | 1595 (0.2) | 48 (0.0) | 523 (0.1) | 475 (0.1) |
| Scenario 1c | The pregnancy occurred before the patient was registered at the practice or before UTS. | 415 807 (44.6) | 204 176 (21.9) | 60 423 (14.2) | 0 (0.0) |
| *Problem 2: The women was pregnant at the time of the database record, but the pregnancy was still ongoing at the end of available follow-up in the database.* | | | | | |
| Scenario 2a | The patient transferred out or died before the putative end of pregnancy. | 177 557 (19.0) | 40 191 (4.3) | 71 012 (16.7) | 117 571 (12.6) |
| Scenario 2b | The last collection date of the practice was before the putative end of pregnancy. | 65 141 (7.0) | 22 039 (2.4) | 24 091 (5.7) | 58 698 (6.3) |
| *Problem 3: The patient was not pregnant at the time of the database record.* | | | | | |
| Scenario 3a | Episode is derived from historical pregnancy information recorded in the first few months after the patient joined the practice. | 10 235 (1.1) | 588 (0.1) | 3058 (0.7) | 3875 (0.4) |
| Scenario 3b | Patient asks for advice while planning a pregnancy. | 0 (0.0) | 0 (0.0) | 0 (0.0) | 0 (0.0) |
| *Problem 4: The pregnancy record belongs to another pregnancy episode in the Register.* | | | | | |
| Scenario 4a | Delay in recording the outcome of a pregnancy, algorithm calculates the last menstrual period date (LMP) too late and uncovers records at the beginning of pregnancy creating this pregnancy with recorded outcome missing. | 61 662 (6.6) | 9299 (1.0) | 23 099 (5.4) | 35 255 (3.8) |
| Scenario 4b | The LMP is derived from the data and is wrong resulting in early codes being uncovered creating this episode. | 29 057 (3.1) | 4022 (0.4) | 11 304 (2.7) | 17 110 (1.8) |
| Scenario 4c | The LMP has been shifted earlier in time uncovering records at the end of the pregnancy. | 0 (0.0) | 0 (0.0) | 0 (0.0) | 0 (0.0) |
| Scenario 4d | A code recorded relating to the patient's delivery history is incorrectly identified by the algorithm as a delivery uncovering records at the end. | 349 874 (37.5) | 113 688 (12.2) | 90 274 (21.3) | 219 505 (23.5) |
| Scenario 4e | The outcome of the pregnancy episode has been misclassified as antenatal | 38 848 (4.2) | 8000 (0.9) | 6611 (1.6) | 18 222 (2.0) |
| None | These pregnancy episodes did not meet the criteria for any identified scenarios. | 106 458 (11.4) | – | – | 94 769 (10.2) |

Continued

| Scenario | Description | N of pregnancy episodes with evidence of this scenario (% of total episodes with missing outcome) | N of pregnancy episodes with evidence of this scenario only (% of total episodes with missing outcome) | N of pregnancy episodes with evidence of an outcome in linked HES (% of linkage eligible episodes with recorded outcome missing†) | N of episodes during current registration and UTS follow-up (% of total episodes with missing outcome)* |
|---|---|---|---|---|---|

**Table 4** Continued

*A version of this table restricted to episodes which occurred during practice UTS follow-up and patient's current registration is given in the appendices (online supplemental appendix 16).
†Denominator=pregnancy episodes which had at least 1-day overlap with the available HES follow-up period and where the woman was eligible for linkage.
CPRD, Clinical Practice Research Datalink; HES, Hospital Episode Statistics; UTS, up-to-standard.

have been too long leading to an overlap (scenario 3a and 3b).

Ten per cent of conflicting pairs had a loss and delivery recorded on the same date and no 'current pregnancy' antenatal codes suggesting they may have been recorded as part of an obstetric history (scenario 2a). Only small percentages of episodes were consistent with other scenarios. Proportional distribution of the scenarios was similar when restricted to those recorded during UTS and current registration to that of the whole Pregnancy Register.

## DISCUSSION

This work has shown that uncertain pregnancy episodes in the CPRD Pregnancy Register can contain valuable information about a woman's pregnancy. A high proportion of the uncertain episodes were during research quality follow-up time and therefore comprise data which would usually be included in study designs.[9] We have systematically identified potential reasons for the existence of uncertain episodes within the pregnancy register to allow researchers to consider in more detail whether inclusion is appropriate for their study. This work adds further value to the CPRD Pregnancy Register which is already unique in its inclusion of all pregnancy data regardless of completion.[3 4] To our knowledge, no previous studies have attempted to examine uncertain pregnancies in EHR data in this way and many of the scenarios we have described will also be applicable to other EHR data sources.

We found that most episodes with a missing outcome could be explained by the outcomes not being captured in the CPRD GOLD primary care database; either the patient was not registered at the time of the pregnancy, the outcome was not recorded by the GP but could be found in linked data, or follow-up ended before the outcome. These are likely to be genuine and contemporaneous pregnancies which would be missed if episodes with recorded outcome missing were excluded from the Register. In fact, most of the scenarios we identified are consistent with the episodes being true and current pregnancies. When conducting drug utilisation or vaccine uptake studies, researchers may wish to include episodes where the database follow-up ended before the outcome to avoid underestimation especially for new drugs or vaccination programmes. Further to our objective to provide guidance, table 6 outlines potential considerations for researchers deciding whether to include or exclude uncertain episodes from their study.

There is evidence to suggest that historical outcomes being recorded by the GP during an ongoing pregnancy may explain a sizeable proportion of the uncertain episodes generated by the algorithm. This can lead to true pregnancies being split by the algorithm and depending on the timing, this will either generate an additional episode with outcome missing or two separate episodes with outcomes (figure 1, step 3). In either case, the resulting episodes may conflict with one another. Based on our findings, this appears to be something that happens fairly frequently. One concern is that these episodes are likely to appear more frequently for women with a history of complicated pregnancy outcomes. For example, previous caesarean sections may be likely to be noted by the GP during current care as would outcomes such as ectopic pregnancies. Researchers should be aware that exclusion of women who have overlapping pregnancies for this reason might therefore systematically exclude those with a history of pregnancy complications, introducing bias.

It is also possible that current pregnancies with serious complications are more likely to have an uncertain episode in the Register. For example, women with pre-eclampsia are more likely to have consultant-led antenatal care carried out in hospital, increasing the chances that their primary care record is incomplete and has no recorded outcome.[10] This data pattern is likely to result in the pregnancy being split into multiple episodes without outcome (figure 1, step 8). Dropping all uncertain episodes at the study design stage may mean that these patients are missed. Researchers who are interested in specific pregnancy complications should take this into consideration and use a tailored approach when selecting a study population.

While some conflicting episodes may be caused by poor quality data, there are many conflicting episodes for which it may be possible to clarify which time period is likely to be the true pregnancy. We found that episode conflicts were more likely to occur for pregnancies ending in loss; this is of little surprise given the wider variation around the true gestation of such pregnancies.[11] There

**Table 5**  Numbers of conflicting pregnancy episodes which were consistent with applied criteria for each scenario*

| Scenario | Description | N of pregnancy pairs with evidence of this scenario (% of total conflicting pregnancy pairs) | N of pairs with evidence of only this scenario (% of total conflicting pregnancy pairs) | N of pairs with a linked baby in the MBL (% of total conflicting pregnancy pairs) | N of pairs with evidence of pregnancy in linked HES (% of pairs eligible for HES linkage†) | N of pairs during current registration and UTS follow-up MBL (% of total conflicting pregnancy pairs) |
|---|---|---|---|---|---|---|
| Denominator | | 251 026 | 251 026 | 251 026 | 160 461† | 251 026 |
| *Problem 1: Both pregnancies are true but one is a current pregnancy and one is a historical pregnancy.* | | | | | | |
| Scenario 1a | The GP records a past delivery or loss during a current pregnancy with the same outcome resulting in another episode being created. | 2464 (1.0) | 413 (0.2) | 2164 (0.9) | 2332 (1.5) | 1981 (0.8) |
| Scenario 1b | A patient has a record relating to a loss recorded during a pregnancy ending in delivery or vice-versa. Conflicting episodes are generated by the algorithm. | 73 208 (29.2) | 35 026 (14.0) | 11 388 (4.5) | 19 900 (12.4) | 31 526 (12.6) |
| *Problem 2: Both pregnancies are historical.* | | | | | | |
| Scenario 2a | A patient has information on historical pregnancies recorded with the current date rather than the actual date. | 27 250 (10.9) | 0 (0.0) | 175 (0.1) | 6835 (4.3) | 12 557 (5.0) |
| *Problem 3: Both pregnancies are true and current but the gestation of the second pregnancy estimated by the algorithm is too long.* | | | | | | |
| Scenario 3a | The woman has two losses which are >8 weeks and <12 weeks apart. | 6425 (2.6) | 12 (0.0) | 0 (0.0) | 1336 (0.8) | 2284 (0.9) |
| Scenario 3b | The woman has two pregnancies close together and the second ends in delivery. If the last menstrual period date (LMP) information is wrong for this pregnancy, then algorithm episodes may overlap. | 32 948 (13.1) | 3705 (1.5) | 1564 (0.6) | 7833 (4.9) | 13 464 (5.4) |
| *Problem 4: The pregnancy is real but is split into separate episodes by the rules of the algorithm.* | | | | | | |
| Scenario 4a | The GP records further information about a pregnancy outcome >25 weeks later for deliveries or >8 weeks or <12 weeks later for losses. | 2939 (1.2) | 251 (0.1) | 2646 (1.1) | 2824 (1.8) | 2347 (0.9) |
| Scenario 4b | The GP records further antenatal information after the end of a pregnancy. Conflicting episodes are generated by the algorithm | 43 581 (17.4) | 40 928 (16.3) | 13 531 (5.4) | 16 718 (10.4) | 27 131 (10.8) |
| Scenario 4c | The patient has a follow-up scan after a pregnancy loss. The scan is recorded in the data as an antenatal scan, a conflicting episode is then generated by the algorithm. | 2734 (1.1) | 0 (0.0) | 0 (0.0) | 744 (0.5) | 2088 (0.8) |
| Scenario 4d | The GP records information about a pregnancy but no outcome with >6 weeks between records. If the second episode has gestational information, the start may be assigned before the start of the first episode. | 14 695 (5.9) | 14 695 (5.9) | 0 (0.0) | 7392 (4.6) | 9911 (3.9) |
| Scenario 4e | The pregnancy dates have been shifted backwards by the rules of the algorithm leaving uncovered records. Conflicting episodes are generated by the algorithm. | 73 191 (29.2) | 70 472 (28.1) | 53 464 (21.3) | 42 785 (26.7) | 55 205 (22.0) |
| None | These pairs of pregnancies did not meet the criteria for any identified scenarios. | 35 449 (14.1) | – | 13 241 (5.3) | 14 173 (8.8) | 15 650 (6.2) |

*A version of this table restricted to episodes which occurred during practice UTS follow-up and patient's current registration is given in the appendices (online supplemental appendix 17).
†Denominator=pregnancy episodes which had at least 1-day overlap with the available HES follow-up period and where the woman was eligible for linkage.
GP, general practitioner; HES, Hospital Episode Statistics; MBL, Mother-Baby-Link; UTS, up-to-standard.

**Table 6** Issues with different approaches to dealing with uncertain episodes and recommendations

| Example uses | Issues with a highly specific approach: excluding all uncertain episodes | Issues with a highly sensitive approach: including all uncertain episodes | Recommended tailored approach: including or excluding uncertain episodes based on scenario criteria |
|---|---|---|---|
| Vaccine uptake study | ► Underestimate of uptake during pregnancy | ► Overestimate of uptake during pregnancy where historical episodes are included | ► Consider using episodes without recorded outcome which continue after data follow-up to maximise the capture of exposure events.<br>► Consider using linked data to obtain additional outcomes.<br>► Exclude episodes which are likely to be derived from historical data based on our described scenarios. |
| Drug/vaccine safety study | ► Underestimation of pregnancies ending in loss<br>► Underestimation of pregnancy complications | ► Misclassification of exposure status<br>► Overestimation of outcomes | ► Consider using linked data to obtain additional outcomes restricting the study population to those patients eligible for linkage.<br>► Exclude episodes which are likely to be derived from historical data based on our described scenarios.<br>► Consider merging conflicting episodes which are consistent with problem 4 and adjusting the timing accordingly (deciding which of the outcomes is likely to be the true outcome based on the scenarios we have described and then estimating a start date. This should be based on a combination of the patient's antenatal records and default duration dependent on outcome type[3]).<br>► Consider ensuring pregnancy start is at least 9 months before the last data collection date to allow for attainment of outcomes. |
| Ascertaining pregnancy history | ► Underestimation of parity<br>► Underestimation of certain pregnancy events<br>► Underestimation of pregnancies ending in loss | ► Overestimation of parity | ► Consider using linked data to obtain additional outcomes restricting the study population to those patients eligible for linkage.<br>► Exclude episodes which are likely to be derived from historical data based on our described scenarios.<br>► Consider ensuring pregnancy start is at least 9 months before the last data collection date to allow for attainment of outcomes. |
| Excluding pregnant women from a study cohort | ► Reduction in potential study population | ► Potential misclassification of pregnancy status<br>► Potential errors in pregnancy timing | ► Consider merging conflicting episodes which are consistent with problem 4 and adjusting the timing accordingly (deciding which of the outcomes is likely to be the true outcome based on the scenarios we have described and then estimating a start date. This should be based on a combination of the patient's antenatal records and a default duration dependent on outcome type[3]).<br>► Consider using linked data to obtain additional outcomes, restricting the study population to those patients eligible for linkage.<br>► Exclude episodes which are likely to be derived from historical data based on our described scenarios. |

was also a large overlap between the conflicting episodes and those that were missing an outcome. Again, this is not surprising as the start and end dates for the missing outcome episodes have large margins of error, given they are often estimated based on one or two antenatal codes (figure 1, step 8).[3] Not including uncertain episodes may lead to underascertainment of miscarriage as an outcome. However, including them all may lead to exposure status misclassification due to mistimed start and end dates or past pregnancy outcomes being counted.

Researchers may consider using multiple imputation to handle missing outcomes. However, there is a strong likelihood that the pattern of missing pregnancy outcomes is not missing at random and both multiple imputation and listwise deletion could result in biased results. Investigation of the linked HES data has shown that using these additional data alongside the Register could help users to identify many missing outcomes.[7 8 12] Potentially useful pregnancy outcome data were found in multiple places across the HES APC database (NHS Digital, 2021). Identifying outcomes in HES could allow users of the Register to adjust the dates of the pregnancy episodes. While HES data are useful as a complementary source

of information, it is also an EHR database derived from data that were not collected for research purposes and there may be gaps in recording. It is, however, less likely that pregnancy outcome events which happen in hospital will be recorded retrospectively and therefore dates of recorded outcomes may be considered more reliable.

Furthermore, using the HES DID data to access antenatal scan records offers a useful way to validate the dates of primary care pregnancy episodes as patients are unlikely to have an antenatal scan when they are not currently pregnant.[13] When using linked data, we recommend that the study population be restricted to those patients in the Pregnancy Register who are eligible for linkage.

The main limitation of this work is that it relies on the assumption that real-life scenarios will consistently result in the same data patterns. EHR data such as CPRD GOLD are not collected for the purposes of research and can be messy for a variety of reasons. As the criteria we applied to identify our proposed scenarios may not have been a true fit to each pregnancy episode, this may have resulted in misclassification of the true underlying cause. While we did validate a random sample of pregnancy episodes by looking at the individual Read codes recorded, it was

not possible to look at every episode in detail. Furthermore, some of our scenarios relied on assumptions as to why and when GPs may record clinical information relating to pregnancy. While this was informed by clinician advice and clinical guidelines, it may not be correct in every case. There is also the possibility that there are other scenarios which we did not identify, and special cases of scenarios that we could not test. For example, since 2007, women in the UK have been given the option of accessing midwife-led care directly. While information about the pregnancy should be fed to their GP, this may not always be the case. A survey report by the Quality Care Commission published in 2020 estimated that in 2018, 47% of women accessed antenatal care directly through a midwife.[14] As yet, no routinely linked data allow for investigation of this special case of scenario 1a.

We have described in detail reasons why uncertain pregnancy episodes may occur in the CPRD Pregnancy Register and criteria which researchers can apply to ascertain which episodes may fit each scenario. This work offers researchers the opportunity to tailor their study to accommodate these episodes where appropriate (table 6).

## CONCLUSIONS

This work has shown evidence that most uncertain pregnancy episodes are consistent with true and current pregnancies for which the data contain valuable information. It is important that researchers carefully consider the impact of including or excluding these episodes from their study. We have demonstrated that examining patterns of events within the primary care data or looking for further evidence in linked data can help to identify possible explanations. Here we offer users of the Pregnancy Register an insight into why these episodes exist and guidance on how to tailor their study population accordingly.

**Acknowledgements** This work uses data provided by patients and collected by the NHS as part of their care and support.

**Contributors** JC, KB, ST, RW, HIM and CM contributed to the initiation, planning and design of the study. JC performed the analysis. KB, ST, RW and CM conducted study supervision. HIM and ST provided clinical input. JC wrote the manuscript with KB, ST, RW, HIM and CM performing critical revision. JC is acting as guarantor for this work

**Funding** This work forms part of JC's PhD which is funded by CPRD (grant number N/A). KB is funded by a Wellcome Senior Research Fellowship (220283/Z/20/Z). CM was supported by a UKRI Innovation Fellowship at Health Data Research UK London (MR/S003932/1). HIM and ST were funded by the National Institute for Health Research (NIHR) Health Protection Research Unit (HPRU) in Immunisation (IS-HPU1112-10096) at the London School of Hygiene and Tropical Medicine in partnership with Public Health England (PHE).

**Disclaimer** The views expressed are those of the authors and not necessarily those of the NHS, the NIHR, the Department of Health and Social Care, or PHE.

**Competing interests** JC and RW are employees of CPRD. There are no other conflicts of interest to report.

**Patient consent for publication** Not required.

**Ethics approval** This study involves human participants and was approved by the Independent Scientific Advisory Committee (ISAC) for Medicines and Healthcare Products Regulatory Agency Database Research (protocol no: 17_285R2

and 19_140) and the London School of Hygiene and Tropical Medicine Ethics Committee. This study uses de-identified electronic health records only.

**Provenance and peer review** Not commissioned; externally peer reviewed.

**Data availability statement** Data may be obtained from a third party and are not publicly available. The data used for this study were obtained from the Clinical Practice Research Datalink (CPRD). All data are available via an application to CPRD's Research Data Governance (RDG) Process (see https://www.cprd.com/research-applications). Data acquisition is associated with a fee and subject to ethics approval.

**ORCID iDs**
Jennifer Campbell http://orcid.org/0000-0002-0684-4437
Helen I McDonald http://orcid.org/0000-0003-0576-2015

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
