## [Reviewer comments · BMJ Open]

ARTICLE DETAILS

TITLE (PROVISIONAL)	Investigating the Optimal Handling of Uncertain Pregnancy Episodes in the CPRD GOLD Pregnancy Register: a methodological study using UK primary care data
AUTHORS	Campbell, Jennifer; Bhaskaran, Krishnan; Thomas, Sara; Williams, Rachael; McDonald, Helen; Minassian, Caroline

VERSION 1 – REVIEW

REVIEWER	Filion, Kristian McGill University, Medicine
REVIEW RETURNED	28-Sep-2021

GENERAL COMMENTS	GENERAL COMMENTS: I thank the authors for the opportunity to review their manuscript, which investigates the optimal handling of uncertain pregnancy episodes in the CPRD Pregnancy Register. The manuscript focuses on pregnancies identified by the algorithm used to construct the CPRD Pregnancy Register that either have missing outcome data or appear to overlap in time with other pregnancy records. This study has many strengths. It examines a methodologically important question as these issues represent important challenges to conducting pregnancy-related research using this data source. Although this manuscript is particularly targeted towards users of the CPRD Gold and its Pregnancy Register, this data source is among the most frequently used for epidemiologic research and it is increasingly being used for pregnancy-related studies since the development of its pregnancy register. The inclusion of the relevant diagnostic and procedure codes in the appendices also increases the potential impact of this work. It does have some potential limitations and other aspects that require clarification; these issues are discussed in the Specific Comments below. SPECIFIC COMMENTS: 1. The majority of the analyses are not restricted to research quality follow-up time. While some analyses restricted to research quality follow-up time are presented in the appendices, some discussion of these results should be added to the main text as this represents the population most likely to be included drug safety studies. 2. HES and other linked data were used to clarify some of the uncertain pregnancy episodes. Some discussion of whether these data should be considered complementary data to the CPRD Pregnancy Register or gold standard data may be helpful. While these data are useful to identifying missing outcomes, do the authors have any suggestions on how to handle situations where
---

	the pregnancy outcomes recorded in these data sources are discordant? 3. Along those lines, if using linked data, do the authors recommend restricting the study population to linkable patients or supplementing CPRD data with linked data for those for whom it is available? The former will ensure that the same data are available for all patients, while the latter will maximize sample size. 4. While a detailed description of the algorithm used to estimate gestational age goes beyond the scope of this manuscript, some discussion of it is needed given its importance in determining overlapping pregnancies and in defining pregnancy outcomes. 5. The authors do not want to be prescriptive in how to handle the issues discussed, but greater guidance is needed in some places of the Discussion. Along those lines, Table 5 is very informative, but a little more detail regarding the recommended tailored approach is warranted. For example, can the authors be more specific about how to adjust “the timing accordingly”? 6. When dealing with missing outcomes, I wonder if it is also worth recommending that the study period be defined in such a way that included pregnancies all have sufficient time to reach the end of pregnancy. In addition to reducing the amount of missing outcome data, it would avoid potential bias by ensuring that all patients included near the end of the study period have the opportunity to have their pregnancy outcome observed rather than just those that end in pregnancy loss. 7. It may be useful to add some discussion of the use of multiple imputation to impute missing outcome data (rather than simply including vs excluding these observations). 8. One reason for incomplete pregnancy data in the CPRD may be that women were closely followed by another healthcare provider (e.g., obstetrician, midwife) and reporting back to the general practitioner was incomplete. While HES data may be useful for women seen by obstetricians, these data will be less informative for those followed by midwives. It is also why we do not anticipate all births to be recorded in HES. To contextualize this issue for the international reader, some discussion of 1) what proportion of births are expected to be recorded in HES; and 2) what proportion of pregnant women are followed by the different healthcare providers in the UK may be helpful. 9. Have there been previous studies examining this issue in other data sources? If so, it may be worth describing them briefly in the Discussion. Can the authors mention briefly if there are lessons that researchers using other data sources can take from this work? 10. Minor points:  a. Title: Please specify that it is the CPRD Gold Pregnancy Register. b. Abstract: Please report the dates of the study period in the Participants section. c. Methods: It would be helpful to report the dates of data availability for the various linked data sources. d. One reason why historical data may be incorrectly believed to be current data is the default dates in the CPRD, which are the
--	---

	data of entry. This should be stated explicitly to help contextualize this issue. e. The inclusion of relevant codes in the appendices strengthens this work and will facilitate update of the study results by researchers working in this area. Did the authors consider also uploading these codes as either text files or Excel files rather than PDFs to facilitate their use by end-users?
--	--

REVIEWER	Nawab, Tabassum Jawaharlal Nehru Medical College, Department of Community Medicine
REVIEW RETURNED	07-Oct-2021

GENERAL COMMENTS	I congratulate the authors for a very meticulous and detailed work on a very important aspect of research conducted using electronic health records! I think they have captured all the scenarios possible, and have done an excellent job of describing the process with apt examples. The discussions and recommendations for tailored approach are very appropriate.
---

REVIEWER	Sultana, MT Dhaka Medical College and Hospital, Obstetrics and gynecology
REVIEW RETURNED	10-Oct-2021

GENERAL COMMENTS	An elaborate study. If the following things considered then it will be more helpful for health recording system.  1. The study design should be more understandable. 2. Study did not fulfill specific objective no 3. 3. There are some minor spelling mistakes  A) page 5 line 35 ...at least one pregnancy B) page 6 line 14 interactive process 4. There should be some proposal about appropriate handling of EHR and also some recommendations about overcome the information gaps in EHR.
--

VERSION 1 – AUTHOR RESPONSE

Response to Reviewer Comments

Responses below in blue:

Reviewer: 1

Dr. Kristian Filion, McGill University

Comments to the Author:

GENERAL COMMENTS:

I thank the authors for the opportunity to review their manuscript, which investigates the optimal handling of uncertain pregnancy episodes in the CPRD Pregnancy Register. The manuscript focuses on pregnancies identified by the algorithm used to construct the CPRD Pregnancy Register that either have missing outcome data or appear to overlap in time with other pregnancy records. This study has many strengths. It examines a methodologically important question as these issues represent important challenges to conducting pregnancy-related research using this data source. Although this manuscript is particularly targeted towards users of the CPRD Gold and its Pregnancy Register, this data source is among the most frequently used for epidemiologic research and it is increasingly being used for pregnancy-related studies since the development of its pregnancy register. The inclusion of the relevant diagnostic and procedure codes in the appendices also increases the potential impact of

this work. It does have some potential limitations and other aspects that require clarification; these issues are discussed in the Specific Comments below.

Thank-you for taking the time to review this manuscript thoroughly. We have found your feedback very useful in revising the paper. We have added responses to each of your specific comments below.

SPECIFIC COMMENTS:

1. The majority of the analyses are not restricted to research quality follow-up time. While some analyses restricted to research quality follow-up time are presented in the appendices, some discussion of these results should be added to the main text as this represents the population most likely to be included drug safety studies.

We have added a discussion of these results to the Results section. The distribution of scenarios was similar to that of the whole Register when restricted to UTS follow-up and current registration.

Changes:

“The distribution of scenarios that occurred during the period left censored by the practice UTS date and patient current registration date was similar to that of the Pregnancy Register as a whole (Appendix 16, Table 3).” (Results P20)

“Proportional distribution of the scenarios was similar when restricted to those recorded during UTS and current registration to that of the whole Pregnancy Register.” (Results P25)

2. HES and other linked data were used to clarify some of the uncertain pregnancy episodes. Some discussion of whether these data should be considered complementary data to the CPRD Pregnancy Register or gold standard data may be helpful. While these data are useful to identifying missing outcomes, do the authors have any suggestions on how to handle situations where the pregnancy outcomes recorded in these data sources are discordant?

We have added a discussion of this point to the Discussion section. We consider that HES data is likely to be more robust in terms of pregnancy outcome dates as it is being recorded at point of care although, as with all EHR data, it needs to be treated with caution and with the context of why it was created in mind.

Changes:

“Whilst HES data is useful as a complementary source of information it is also an EHR database derived from data that were not collected for research purposes and there may be gaps in recording. It is however, less likely that pregnancy outcome events which happen in hospital will be recorded retrospectively and therefore dates of recorded outcomes may be considered more reliable.” (Discussion P30)

3. Along those lines, if using linked data, do the authors recommend restricting the study population to linkable patients or supplementing CPRD data with linked data for those for whom it is available? The former will ensure that the same data are available for all patients, while the latter will maximize sample size.

We recommend restricting the study population to linkage eligible patients when utilising linked data to define study outcomes to ensure that the study population is comparable- we have added this to the Discussion.

Changes:

“When utilising linked data to we recommend that the study population is restricted to those patients in the Pregnancy Register who are eligible for linkage.” (Discussion P30)

4. While a detailed description of the algorithm used to estimate gestational age goes beyond the scope of this manuscript, some discussion of it is needed given its importance in determining overlapping pregnancies and in defining pregnancy outcomes.

We have included all the relevant gestational information in Figure 1 which outlines the algorithm and have now edited the manuscript to refer readers to the relevant section of this figure whenever we discuss the algorithm. We agree a more detailed description of the algorithm used to estimate gestational age goes beyond the scope of this manuscript.

5. The authors do not want to be prescriptive in how to handle the issues discussed, but greater guidance is needed in some places of the Discussion. Along those lines, Table 5 is very informative, but a little more detail regarding the recommended tailored approach is warranted. For example, can the authors be more specific about how to adjust “the timing accordingly”?

We have added more detail to Table 5.

Changes:

(Table 5 P32-34)

- *“Consider utilising episodes without recorded outcome which continue after data follow-up to maximise the capture of exposure events.”*
- *“Consider merging conflicting episodes which are consistent with problem 4 and adjusting the timing accordingly (deciding which of the outcomes is likely to be the true outcome based on the scenarios we have described and then estimating a start date. This should be based on a combination of the patient’s antenatal records and default duration dependent on outcome type (3)).”*

6. When dealing with missing outcomes, I wonder if it is also worth recommending that the study period be defined in such a way that included pregnancies all have sufficient time to reach the end of pregnancy. In addition to reducing the amount of missing outcome data, it would avoid potential bias by ensuring that all patients included near the end of the study period have the opportunity to have their pregnancy outcome observed rather than just those that end in pregnancy loss.

We have added guidance on this point to Table 5.

Changes:

(Table 5 P32-34)

- *“Consider ensuring pregnancy start is at least nine months before the last data collection date to allow for attainment of outcomes.”*

7. It may be useful to add some discussion of the use of multiple imputation to impute missing outcome data (rather than simply including vs excluding these observations).

We have added guidance on the use of multiple imputation to the Discussion section. Given that pregnancy outcomes are unlikely to be missing at random we recommend the use of linked data rather than multiple imputation.

Changes:

“Researchers may consider using multiple imputation to handle missing outcomes. However, there is a strong likelihood that the pattern of missing pregnancy outcomes is not missing at random and both multiple imputation and listwise deletion could result in biased results.” (Discussion P30)

8. One reason for incomplete pregnancy data in the CPRD may be that women were closely followed by another healthcare provider (e.g., obstetrician, midwife) and reporting back to the general practitioner was incomplete. While HES data may be useful for women seen by obstetricians, these data will be less informative for those followed by midwives. It is also why we do not anticipate all births to be recorded in HES. To contextualize this issue for the international reader, some discussion of 1) what proportion of births are expected to be recorded in HES; and 2) what proportion of pregnant women are followed by the different healthcare providers in the UK may be helpful.

We agree that HES data may be less informative for women who receive midwife led care, especially if they deliver somewhere other than a hospital. We have added information on this special case of scenario 1a to the Discussion. We raised this point with a GP colleague, and we understand that midwife led care should be fed back to the woman's GP and added to their primary care record. However, we recognise that this may not always be the case.

Changes:

"There is also the possibility that there are other scenarios which we did not identify, and special cases of scenarios that we could not test. For example, since 2007 women in the UK have been given the option of accessing midwife led care directly. Whilst information about the pregnancy should be fed to their GP this may not always be the case. A survey report by the Quality Care Commission published in 2020 estimated that in 2018 47% of women accessed antenatal care directly through a midwife (14). As yet, no routinely linked data allow for investigation of this special case of Scenario 1a." (Discussion P30)

9. Have there been previous studies examining this issue in other data sources? If so, it may be worth describing them briefly in the Discussion. Can the authors mention briefly if there are lessons that researchers using other data sources can take from this work?

We have reviewed the literature but not identify any relevant papers which have looked at this issue. All previous Pregnancy Registers in EHR data have taken the approach of dropping less certain records when generating the register. We have now included a statement to this effect in the paper.

Changes:

"To our knowledge no previous studies have attempted to examine uncertain pregnancies in EHR data in this way and many of the scenarios we have described will also be applicable to other EHR data sources." (Discussion P29)

10. Minor points:

a. Title: Please specify that it is the CPRD Gold Pregnancy Register.

We have added this.

b. Abstract: Please report the dates of the study period in the Participants section.

We have added this

c. Methods: It would be helpful to report the dates of data availability for the various linked data sources.

We have added this

d. One reason why historical data may be incorrectly believed to be current data is the default dates in the CPRD, which are the data of entry. This should be stated explicitly to help contextualize this issue. We have added this to scenario 3a to make it more explicit to the reader.

e. The inclusion of relevant codes in the appendices strengthens this work and will facilitate update of the study results by researchers working in this area. Did the authors consider also uploading these codes as either text files or Excel files rather than PDFs to facilitate their use by end-users?

Unfortunately, this was not an option upon submission of the manuscript. However, we would be happy to provide these to end-users if they contact the corresponding author.

Reviewer: 2

Dr. Tabassum Nawab, Jawaharlal Nehru Medical College

Comments to the Author:

I congratulate the authors for a very meticulous and detailed work on a very important aspect of research conducted using electronic health records! I think they have captured all the scenarios possible and have done an excellent job of describing the process with apt examples. The discussions and recommendations for tailored approach are very appropriate.

Thank-you for your kind comments. We are pleased that you enjoyed the paper.

Reviewer: 3

Dr. MT Sultana, Dhaka Medical College and Hospital, Ibn Sina Medical College

Comments to the Author:

An elaborate study. If the following things considered, then it will be more helpful for health recording system.

Thank-you for taking the time to review this manuscript. We have taken on board your feedback and responded to your specific comments below.

1. The study design should be more understandable.

We appreciate that this is not a traditional epidemiological study with a familiar design, but rather a methodological paper which aims to give advice to other researchers when designing their studies. We have been as detailed as possible about the way in which we identified scenarios which may result in uncertain pregnancy episodes and the criteria which we applied to the data in order to look for evidence of each scenario. However, if there are any specific design issues which are unclear, we are happy to provide further clarification.

2. Study did not fulfill specific objective no 3.

We apologise that the fulfilment of objective 3 was not clear. Additional detail has been added throughout the paper, and in Table 5 to clarify this. In our discussion and in particularly Table 5 we have contextualised our results and provided guidance for users of the Pregnancy Register as to how to handle uncertain episodes when designing studies.

3. There are some minor spelling mistakes

A) page 5 line 35 ...at least one pregnancy

We have corrected this.

B) page 6 line 14 interactive process

No change made – the intention is for this to read iterative (i.e. repeating a process again and again, each time applying it to the result of the previous stage).

4. There should be some proposal about appropriate handling of EHR and also some recommendations about overcome the information gaps in EHR.

One of the key aims of the paper was to provide information to inform the handling of EHR data relating to pregnancy, and we summarise our recommendations in Table 5. We have added further clarifications to this table further to reviewer 1 comment 5. A key finding from our analysis of linked data was that hospital records could help to overcome some of the information gaps (e.g. missing outcomes) in the primary care record, as discussed in the paper. Further to reviewer 1 comment 7, we have also added further detail to the Discussion section in this revision, including specific guidance on handling missing data.

VERSION 2 – REVIEW

REVIEWER	Filion, Kristian McGill University, Medicine
REVIEW RETURNED	13-Dec-2021

GENERAL COMMENTS	I thank the authors for the opportunity to review their manuscript, which is a resubmission of a methodological paper that describes the optimal handling of uncertain pregnancy episodes in the CPRD Gold Pregnancy Register. The authors have incorporated the comments raised during the previous review, which has further strengthened this work. I have no additional comments regarding their manuscript.
--